# Best Arm Identification for Stochastic Rising Bandits

## Abstract

Stochastic Rising Bandits (SRBs) model sequential decision-making problems in which the expected reward of the available options increases every time they are selected. This setting captures a wide range of scenarios in which the available options are *learning entities* whose performance improves (in expectation) over time. While previous works addressed the regret minimization problem, this paper focuses on the *fixed-budget Best Arm Identification* (BAI) problem for SRBs. In this scenario, given a fixed budget of rounds, we are asked to provide a recommendation about the best option at the end of the identification process. We propose two algorithms to tackle the above-mentioned setting, namely R-UCBE, which resorts to a UCB-like approach, and R-SR, which employs a successive reject procedure. Then, we prove that, with a sufficiently large budget, they provide guarantees on the probability of properly identifying the optimal option at the end of the learning process. Furthermore, we derive a lower bound on the error probability, matched by our R-SR (up to logarithmic factors), and illustrate how the need for a sufficiently large budget is unavoidable in the SRB setting. Finally, we numerically validate the proposed algorithms in synthetic and real-world environments and compare them with the currently available BAI strategies.

## 1 Introduction

Multi-Armed Bandits (MAB, Lattimore and Szepesvári, 2020) are a well-known framework that effectively solves learning problems requiring sequential decisions. Given a time horizon, the learner chooses, at each round, a single option (a.k.a. arm) and observes the corresponding noisy reward, which is a realization of an unknown distribution. The MAB problem is commonly studied in two flavours: *regret minimization* (Auer et al., 2002) and *best arm identification* (Bubeck et al., 2009). In regret minimization, the goal is to control the cumulative loss w.r.t. the optimal arm over a time horizon. Conversely, in best arm identification, the goal is to provide a recommendation about the best arm at the end of the time horizon. Specifically, we are interested in the fixed-budget scenario, where we seek to minimize the error probability of recommending the wrong arm at the end of the time budget, no matter the loss incurred during learning.

This work focuses on the *Stochastic Rising Bandits* (SRB), a specific instance of the *rested* bandit (Tekin and Liu, 2012) setting in which the expected reward of an arm increases according to the number of times it has been pulled. Online learning in such a scenario has been recently addressed from a regret minimization perspective by Metelli et al. (2022), in which the authors provide no-regret algorithms for the SRB setting in both the rested and restless cases. The SRB setting models several real-world scenarios where arms improve their performance over time. A classic example is the so-called *Combined Algorithm Selection and Hyperparameter optimization* (CASH, Thornton et al., 2013; Kotthoff et al., 2017; Erickson et al., 2020; Li et al., 2020; Zöller and Huber, 2021), a problem of paramount importance in *Automated Machine Learning* (AutoML, Feurer et al., 2015; Yao et al., 2018; Hutter et al., 2019; Mussi et al., 2023). In CASH, the goal is to identify the *best learning algorithm* together with the *best hyperparameter* configuration for a given ML task (e.g.,

Submitted to 37th Conference on Neural Information Processing Systems (NeurIPS 2023). Do not distribute.

classification or regression). In this problem, every arm represents a hyperparameter tuner acting on a specific learning algorithm. A pull corresponds to a unit of time/computation in which we improve (on average) the hyperparameter configuration (via the tuner) for the corresponding learning algorithm. CASH was handled in a bandit *Best Arm Identification* (BAI) fashion in Li et al. (2020) and Cella et al. (2021). The former handles the problem by considering rising rested bandits with *deterministic* rewards, failing to represent the intrinsic uncertain nature of such processes. Instead, the latter, while allowing stochastic rewards, assumes that the expected rewards evolve according to a *known* parametric functional class, whose parameters have to be learned.[1]

**Original Contributions** In this paper, we address the design of algorithms to solve the BAI task in the rested SRB setting when a *fixed budget* is provided.[2] More specifically, we are interested in algorithms guaranteeing a sufficiently large probability of recommending the arm with the largest expected reward *at the end* of the time budget (as if only this arm were pulled from the beginning). The main contributions of the paper are summarized as follows:[3]

- We propose two *algorithms* to solve the BAI problem in the SRB setting: R-UCBE (an optimistic approach, Section 4) and R-SR (a phases-based rejection algorithm, Section 5). First, we introduce specifically designed estimators required by the algorithms (Section 3). Then, we provide guarantees on the error probability of the misidentification of the best arm.
- We derive the first error probability *lower bound* for the SRB setting, matched by our R-SR algorithm up to logarithmic factors, which highlights the complexity of the problem and the need for a sufficiently large time budget (Section 6).
- Finally, we conduct *numerical simulations* on synthetically generated data and a real-world online best model selection problem. We compare the proposed algorithms with the ones available in the bandit literature to tackle the SRB problem (Section 7).

# 2 Problem Formulation

In this section, we revise the Stochastic Rising Bandits (SRB) setting (Heidari et al., 2016; Metelli et al., 2022). Then, we formulate our best arm identification problem, introduce the definition of error probability, and provide a preliminary characterization of the problem.

**Setting** We consider a rested Multi-Armed Bandit problem $\boldsymbol{\nu} = (\nu_i)_{i \in \llbracket K \rrbracket}$ with a finite number of arms $K$.[4] Let $T \in \mathbb{N}$ be the time budget of the learning process. At every round $t \in \llbracket T \rrbracket$, the agent selects an arm $I_t \in \llbracket K \rrbracket$, plays it, and observes a reward $x_t \sim \nu_{I_t}(N_{I_t,t})$, where $\nu_{I_t}(N_{I_t,t})$ is the reward distribution of the chosen arm $I_t$ at round $t$ and depends on the number of pulls performed so far $N_{i,t} := \sum_{\tau=1}^{t} \mathbb{1}\{I_\tau = i\}$ (i.e., rested). The rewards are stochastic, formally $x_t := \mu_{I_t}(N_{I_t,t}) + \eta_t$, where $\mu_{I_t}(\cdot)$ is the expected reward of arm $I_t$ and $\eta_t$ is a zero-mean $\sigma^2$-subgaussian noise, conditioned to the past.[5] As customary in the bandit literature, we assume that the rewards are bounded in expectation, formally $\mu_i(n) \in [0,1], \forall i \in \llbracket K \rrbracket, n \in \llbracket T \rrbracket$. As in (Metelli et al., 2022), we focus on a particular family of rested bandits in which the expected rewards are monotonically *non-decreasing* and *concave* in expectation.

**Assumption 2.1** (Non-decreasing and concave expected rewards). *Let $\boldsymbol{\nu}$ be a rested MAB, defining $\gamma_i(n) := \mu_i(n+1) - \mu_i(n)$, for every $n \in \mathbb{N}$ and every arm $i \in \llbracket K \rrbracket$ the rewards are non-decreasing and concave, formally:*

$$\textit{Non-decreasing:} \quad \gamma_i(n) \geqslant 0, \qquad\qquad \textit{Concave:} \quad \gamma_i(n+1) \leqslant \gamma_i(n).$$

Intuitively, the $\gamma_i(n)$ represents the *increment* of the real process $\mu_i(\cdot)$ evaluated at the $n^{\text{th}}$ pull. Notice that concavity emerges in several settings, such as the best model selection and economics, representing the decreasing marginal returns (Lehmann et al., 2001; Heidari et al., 2016).

---

[1] A complete discussion of the related works is available in Appendix A. Additional motivating examples are discussed in Appendix B.

[2] We focus on the rested setting only and, thus, from now on, we will omit "rested" in the setting name.

[3] The proofs of all the statements in this work are provided in Appendix D.

[4] Let $y, z \in \mathbb{N}$, we denote with $\llbracket z \rrbracket := \{1, \ldots, z\}$, and with $\llbracket y, z \rrbracket := \{y, \ldots, z\}$.

[5] A zero-mean random variable $x$ is $\sigma^2$-subgaussian if it holds $\mathbb{E}_x[e^{\xi x}] \leqslant e^{\frac{\sigma^2 \xi^2}{2}}$ for every $\xi \in \mathbb{R}$.

**Learning Problem** The goal of BAI in the SRB setting is to select the arm providing the largest expected reward with a large enough probability given a fixed budget $T \in \mathbb{N}$. Unlike the stationary BAI problem (Audibert et al., 2010), in which the optimal arm is not changing, in this setting, we need to decide *when* to evaluate the optimality of an arm. We define optimality by considering the largest expected reward at time $T$. Formally, given a time budget $T$, the optimal arm $i^*(T) \in [\![K]\!]$, which we assume unique, satisfies:

$$i^*(T) := \arg\max_{i \in [\![K]\!]} \mu_i(T),$$

where we highlighted the dependence on $T$ as, with different values of the budget, $i^*(T)$ may change. Let $i \in [\![K]\!] \backslash \{i^*(T)\}$ be a suboptimal arm, we define the suboptimality gap as $\Delta_i(T) := \mu_{i^*(T)}(T) - \mu_i(T)$. We employ the notation $(i) \in [\![K]\!]$ to denote the $i^{\text{th}}$ best arm at time $T$ (arbitrarily breaking ties), i.e., we have $\Delta_{(2)}(T) \leqslant \cdots \leqslant \Delta_{(K)}(T)$. Given an algorithm $\mathfrak{A}$ that recommends $\hat{I}^*(T) \in [\![K]\!]$ at the end of the learning process, we measure its performance with the *error probability*, i.e., the probability of recommending a suboptimal arm at the end of the time budget $T$:

$$e_T(\mathfrak{A}) := \mathbb{P}_{\mathfrak{A}}(\hat{I}^*(T) \neq i^*(T)).$$

**Problem Characterization** We now provide a characterization of a specific class of polynomial functions to upper bound the increments $\gamma_i(n)$.

**Assumption 2.2** (Bounded $\gamma_i(n)$). *Let $\boldsymbol{\nu}$ be a rested MAB, there exist $c > 0$ and $\beta > 1$ such that for every arm $i \in [\![K]\!]$ and number of pulls $n \in [\![0, T]\!]$ it holds that $\gamma_i(n) \leqslant cn^{-\beta}$.*

We anticipate that, even if our algorithms will not require such an assumption, it will be used for deriving the lower bound and for providing more human-readable error probability guarantees. Furthermore, we observe that our Assumption 2.2 is fulfilled by a strict superset of the functions employed in Cella et al. (2021).

# 3 Estimators

In this section, we introduce the estimators of the arm expected reward employed by the proposed algorithms.[6] A visual representation of such estimators is provided in Figure 1.

Let $\varepsilon \in (0, 1/2)$ be the fraction of samples collected up to the current time $t$ we use to build estimators of the expected reward. We employ an *adaptive arm-dependent window* size $h(N_{i,t-1}) := \lfloor \varepsilon N_{i,t-1} \rfloor$ to include the most recent samples collected only, avoiding the use of samples that are no longer representative. We define the set of the last $h(N_{i,t-1})$ rounds in which the $i^{\text{th}}$ arm was pulled as:

$$\mathcal{T}_{i,t} := \{\tau \in [\![T]\!] : I_\tau = i \wedge N_{i,\tau} = N_{i,t-1} - l, \, l \in [\![0, h(N_{i,t-1}) - 1]\!]\}.$$

Furthermore, the set of the pairs of rounds $\tau$ and $\tau'$ belonging to the sets of the last and second-last $h(N_{i,t-1})$-wide windows of the $i^{\text{th}}$ arm is defined as:

$$\mathcal{S}_{i,t} := \big\{(\tau, \tau') \in [\![T]\!] \times [\![T]\!] : I_\tau = I_{\tau'} = i \wedge N_{i,\tau} = N_{i,t-1} - l,$$
$$N_{i,\tau'} = N_{i,\tau} - h(N_{i,t-1}), \, l \in [\![0, h(N_{i,t-1}) - 1]\!]\big\}.$$

In the following, we design a *pessimistic* estimator and an *optimistic* estimator of the expected reward of each arm at the end of the budget time $T$, i.e., $\mu_i(T)$.[7]

**Pessimistic Estimator** The *pessimistic* estimator $\hat{\mu}_i(N_{i,t-1})$ is a negatively biased estimate of $\mu_i(T)$ obtained assuming that the function $\mu_i(\cdot)$ remains constant up to time $T$. This corresponds to the minimum admissible value under Assumption 2.1 (due to the *Non-decreasing* constraint). This estimator is an average of the last $h(N_{i,t-1})$ observed rewards collected from the $i^{\text{th}}$ arm, formally:

$$\hat{\mu}_i(N_{i,t-1}) := \frac{1}{h(N_{i,t-1})} \sum_{\tau \in \mathcal{T}_{i,t}} x_\tau. \tag{1}$$

The estimator enjoys the following concentration property.

---

[6]The estimators are adaptations of those presented by Metelli et al. (2022) to handle a fixed time budget $T$.

[7]Naïvely computing the estimators from their definition requires $\mathcal{O}(h(N_{i,t-1}))$ number of operations. An efficient way to incrementally update them, using $\mathcal{O}(1)$ operations, is provided in Appendix C.

**Lemma 3.1** (Concentration of $\hat{\mu}_i$). *Under Assumption 2.1, for every $a > 0$, simultaneously for every arm $i \in [\![K]\!]$ and number of pulls $n \in [\![0, T]\!]$, with probability at least $1 - 2TKe^{-a/2}$ it holds that:*

$$\hat{\beta}_i(n) - \hat{\zeta}_i(n) \leqslant \hat{\mu}_i(n) - \mu_i(n) \leqslant \hat{\beta}_i(n), \tag{2}$$

*where $\hat{\beta}_i(n) := \sigma\sqrt{\frac{a}{h(n)}}$ and $\hat{\zeta}_i(n) := \frac{1}{2}(2T - n + h(n) - 1)\,\gamma_i(n - h(n) + 1)$.*

As supported by intuition, we observe that the estimator is affected by a negative bias that is represented by $\hat{\zeta}_i(n)$ that vanishes as $n \to \infty$ under Assumption 2.1 with a rate that depends on the increment functions $\gamma_i(\cdot)$. Considering also the term $\hat{\beta}_i(n)$ and recalling that $h(n) = \mathcal{O}(n)$, under Assumption 2.2, the overall concentration rate is $\mathcal{O}(n^{-1/2} + cTn^{-\beta})$.

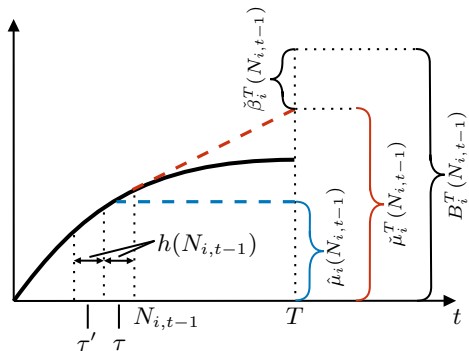

**Optimistic Estimator** The *optimistic* estimator $\check{\mu}_i^T(N_{i,t-1})$ is a positively biased estimation of $\mu_i(T)$ obtained assuming that function $\mu_i(\cdot)$ linearly increases up to time $T$. This corresponds to the maximum value admissible under Assumption 2.1 (due to the *Concavity* constraint). The estimator is constructed by adding to the pessimistic estimator $\hat{\mu}_i(N_{i,t-1})$ an estimate of the increment occurring in the next step up to $T$. The latter uses the last $2h(N_{i,t-1})$ samples to obtain an upper bound of such growth thanks to the concavity assumption, formally:

Figure 1: Graphical representation of the pessimistic $\hat{\mu}_i(N_{i,t-1})$ and the optimistic $\check{\mu}_i^T(N_{i,t-1})$ estimators.

$$\check{\mu}_i^T(N_{i,t-1}) := \hat{\mu}_i(N_{i,t-1}) + \sum_{(j,k) \in \mathcal{S}_{i,t}} (T - j)\frac{x_j - x_k}{h(N_{i,t-1})^2}. \tag{3}$$

The estimator displays the following concentration guarantee.

**Lemma 3.2** (Concentration of $\check{\mu}_i^T$). *Under Assumption 2.1, for every $a > 0$, simultaneously for every arm $i \in [\![K]\!]$ and number of pulls $n \in [\![0, T]\!]$, with probability at least $1 - 2TKe^{-a/10}$ it holds that:*

$$\check{\beta}_i^T(n) \leqslant \check{\mu}_i^T(n) - \mu_i(n) \leqslant \check{\beta}_i^T(n) + \check{\zeta}_i^T(n), \tag{4}$$

*where $\check{\beta}_i^T(n) := \sigma \cdot (T - n + h(n) - 1)\sqrt{\frac{a}{h(n)^3}}$ and $\check{\zeta}_i^T(n) := \frac{1}{2}(2T - n + h(n) - 1)\,\gamma_i(n - 2h(n) + 1)$.*

Differently from the pessimistic estimation, the optimistic one displays a positive vanishing bias $\check{\zeta}_i^T(n)$. Under Assumption 2.2, we observe that the overall concentration rate is $\mathcal{O}(Tn^{-3/2} + cTn^{-\beta})$.

## 4 Optimistic Algorithm: `Rising Upper Confidence Bound Exploration`

In this section, we introduce and analyze `Rising Upper Confidence Bound Exploration` (`R-UCBE`) an *optimistic* error probability minimization algorithm for the SRB setting with a fixed budget. The algorithm explores by means of a UCB-like approach and, for this reason, makes use of the optimistic estimator $\check{\mu}_i^T$ plus a bound to account for the uncertainty of the estimation.[8]

**Algorithm** The algorithm, whose pseudo-code is reported in Algorithm 1, requires as input an exploration parameter $a \geqslant 0$, the window size $\varepsilon \in (0, 1/2)$, the time budget $T$, and the number of arms $K$. At first, it initializes to zero the counters $N_{i,0}$, and sets to $+\infty$ the upper bounds $B_i^T(N_{i,0})$ of all the arms (Line 2). Subsequently, at each time $t \in [\![T]\!]$, the algorithm selects the arm $I_t$ with the largest upper confidence bound (Line 4):

$$I_t \in \arg\max_{i \in [\![K]\!]} B_i^T(N_{i,t-1}) := \check{\mu}_i^T(N_{i,t-1}) + \check{\beta}_i^T(N_{i,t-1}), \tag{5}$$

$$\text{with:} \quad \check{\beta}_i^T(N_{i,t-1}) := \sigma \cdot (T - N_{i,t-1} + h(N_{i,t-1}) - 1)\sqrt{\frac{a}{h(N_{i,t-1})^3}}, \tag{6}$$

---

[8]In `R-UCBE`, the choice of considering the optimistic estimator is natural and obliged since the pessimistic estimator is affected by negative bias and cannot be used to deliver optimistic estimates.

where $\breve{\beta}_i^T(N_{i,t-1})$ represents the exploration bonus (a graphical representation is reported in Figure 1). Once the arm is chosen, the algorithm plays it and observes the feedback $x_t$ (Line 5). Then, the optimistic estimate $\breve{\mu}_{I_t}^T(N_{I_t,t})$ and the exploration bonus $\breve{\beta}_{I_t}^T(N_{I_t,t})$ of the selected arm $I_t$ are updated (Lines 8-9). The procedure is repeated until the algorithm reaches the time budget $T$. The final recommendation of the best arm is performed using the last computed values of the bounds $B_i^T(N_{i,T})$, returning the arm $\hat{I}^*(T)$ corresponding to the largest upper confidence bound (Line 12).

**Bound on the Error Probability of** R-UCBE We now provide bounds on the error probability for R-UCBE. We start with a general analysis that makes no assumption on the increments $\gamma_i(\cdot)$ and, then, we provide a more explicit result under Assumption 2.2. The general result is formalized as follows.

**Theorem 4.1.** *Under Assumption 2.1, let $a^*$ be the largest positive value of $a$ satisfying:*

$$T - \sum_{i \neq i^*(T)} y_i(a) \geqslant 1, \tag{7}$$

*where for every $i \in [\![K]\!]$, $y_i(a)$ is the largest integer for which it holds:*

$$\underbrace{T\gamma_i(\lfloor(1-2\varepsilon)y\rfloor)}_{(A)} + \underbrace{2T\sigma\sqrt{\frac{a}{\lfloor\varepsilon y\rfloor^3}}}_{(B)} \geqslant \Delta_i(T). \tag{8}$$

*If $a^*$ exists, then for every $a \in [0, a^*]$ the error probability of R-UCBE is bounded by:*

$$e_T(\text{R-UCBE}) \leqslant 2TK \exp\left(-\frac{a}{10}\right). \tag{9}$$

Some comments are in order. First, $a^*$ is defined implicitly, depending on the constants $\sigma$, $T$, the increments $\gamma_i(\cdot)$, and the suboptimality gaps $\Delta_i(T)$. In principle, there might exist no $a^* > 0$ fulfilling condition in Equation (7) (this can happen, for instance, when the budget $T$ is not large enough), and, in such a case, we are unable to provide theoretical guarantees on the error probability of R-UCBE. Second, the result presented in Theorem 4.1 holds for generic increasing and concave expected reward functions. This result shows that, as expected, the error probability decreases when the exploration parameter $a$ increases. However, this behavior stops when we reach the threshold $a^*$. Intuitively, the value of $a^*$ sets the maximum amount of exploration we should use for learning.

Under Assumption 2.2, i.e., using the knowledge on the increment $\gamma_i(\cdot)$ upper bound, we derive a result providing conditions on the time budget $T$ under which $a^*$ exists and an explicit value for $a^*$.

**Corollary 4.2.** *Under Assumptions 2.1 and 2.2, if the time budget $T$ satisfies:*

$$T \geqslant \begin{cases} \left(c^{\frac{1}{\beta}}(1-2\varepsilon)^{-1}\left(H_{1,1/\beta}(T)\right) + (K-1)\right)^{\frac{\beta}{\beta-1}} & \text{if } \beta \in (1, 3/2) \\ \left(c^{\frac{2}{3}}(1-2\varepsilon)^{-\frac{2}{3}\beta}\left(H_{1,2/3}(T)\right) + (K-1)\right)^3 & \text{if } \beta \in [3/2, +\infty) \end{cases}, \tag{10}$$

*there exists $a^* > 0$ defined as:*

$$a^* = \begin{cases} \frac{\epsilon^3}{4\sigma^2}\left(\left(\frac{T^{1-1/\beta}-(K-1)}{H_{1,1/\beta}(T)}\right)^\beta - c(1-2\varepsilon)^{-\beta}\right)^2 & \text{if } \beta \in (1, 3/2) \\ \frac{\epsilon^3}{4\sigma^2}\left(\left(\frac{T^{1/3}-(K-1)}{H_{1,2/3}(T)}\right)^{3/2} - c(1-2\varepsilon)^{-\beta}\right)^2 & \text{if } \beta \in [3/2, +\infty) \end{cases},$$

*where $H_{1,\eta}(T) := \sum_{i \neq i^*(T)} \frac{1}{\Delta_i^\eta(T)}$ for $\eta > 0$. Then, for every $a \in [0, a^*]$, the error probability of R-UCBE is bounded by:*

$$e_T(\text{R-UCBE}) \leqslant 2TK \exp\left(-\frac{a}{10}\right).$$

First of all, we notice that the error probability $e_T(\text{R-UCBE})$ presented in Theorem 4.2 holds under the condition that the time budget $T$ fulfills Equation (10). We defer a more detailed discussion on this condition to Remark 5.1, where we show that the existence of a finite value of $T$ fulfilling Equation (10) is ensured under mild conditions.

Let us remark that term $H_{1,\eta}(T)$ characterizes the complexity of the SRB setting, corresponding to term $H_1$ of Audibert et al. (2010) for the classical BAI problem when $\eta = 2$. As expected, in the small-$\beta$ regime (i.e., $\beta \in (1, 3/2)$), looking at the dependence of $H_{1,1/\beta}(T)$ on $\beta$, we realize that

| **Algorithm 1:** R-UCBE. | **Algorithm 2:** R-SR. |
|---|---|
| **Input:** Time budget $T$, Number of arms $K$, Window size $\varepsilon$, Exploration parameter $a$ | **Input:** Time budget $T$, Number of arms $K$, Window size $\varepsilon$ |
| 1 Initialize $N_{i,0} = 0$, | 1 Initialize $t \leftarrow 1$, $N_0 = 0$, $\mathcal{X}_0 = [\![K]\!]$ |
| 2 $\quad B_i^T(0) = +\infty, \forall i \in [\![K]\!]$ | 2 **for** $j \in [\![K-1]\!]$ **do** |
| 3 **for** $t \in [\![T]\!]$ **do** | 3 $\quad$ **for** $i \in \mathcal{X}_{j-1}$ **do** |
| 4 $\quad$ Compute $I_t \in \arg\max_{i \in [\![K]\!]} B_i^T(N_{i,t-1})$ | 4 $\quad\quad$ **for** $l \in [\![N_{j-1}+1, N_j]\!]$ **do** |
| 5 $\quad$ Pull arm $I_t$ and observe $x_t$ | 5 $\quad\quad\quad$ Pull arm $i$ and observe $x_t$ |
| 6 $\quad$ $N_{I_t,t} \leftarrow N_{I_t,t-1} + 1$ | 6 $\quad\quad\quad$ $t \leftarrow t+1$ |
| 7 $\quad$ $N_{i,t} \leftarrow N_{i,t-1}, \quad \forall i \neq I_t$ | 7 $\quad\quad$ **end** |
| 8 $\quad$ Update $\check{\mu}_{I_t}^T(N_{I_t,t})$ | 8 $\quad\quad$ Update $\hat{\mu}_i(N_j)$ |
| 9 $\quad$ Update $\check{\beta}_{I_t}^T(N_{I_t,t})$ | 9 $\quad$ **end** |
| 10 $\quad$ Compute $B_{I_t}^T(N_{I_t,t}) = \check{\mu}_{I_t}^T(N_{I_t,t}) + \check{\beta}_{I_t}^T(N_{I_t,t})$ | 10 $\quad$ Define $\overline{I}_j \in \arg\min_{i \in \mathcal{X}_{j-1}} \hat{\mu}_i(N_j)$ |
| 11 **end** | 11 $\quad$ Update $\mathcal{X}_j = \mathcal{X}_{j-1} \setminus \{\overline{I}_j\}$ |
| 12 Recommend $\widehat{I}^*(T) \in \arg\max_{i \in [\![K]\!]} B_i^T(N_{i,T})$ | 12 **end** |
| | 13 Recommend $\widehat{I}^*(T) \in \mathcal{X}_{K-1}$ (unique) |

179 the complexity of a problem decreases as the parameter $\beta$ increases. Indeed, the larger $\beta$, the faster
180 the expected reward reaches a stationary behavior. Nevertheless, even in the large-$\beta$ regime (i.e.,
181 $\beta > 3/2$), the complexity of the problem is governed by $H_{1,2/3}(T)$, leading to an error probability
182 larger than the corresponding one for BAI in standard bandits (Audibert et al., 2010). This can be
183 explained by the fact that R-UCBE uses the optimistic estimator that, as shown in Section 3, enjoys a
184 slower concentration rate compared to the standard sample mean, even for stationary bandits.

185 This two-regime behavior has an interesting interpretation when comparing Corollary 4.2 with
186 Theorem 4.1. Indeed, $\beta = 3/2$ is the break-even threshold in which the two terms of the l.h.s. of
187 Equation (8) have the same convergence rate. Specifically, the term $(A)$ takes into account the
188 expected rewards growth (i.e., the bias in the estimators), while $(B)$ considers the uncertainty in
189 the estimations of the R-UCBE algorithm (i.e., the variance). Intuitively, when the expected reward
190 function displays a slow growth (i.e., $\gamma_i(n) \leqslant cn^{-\beta}$ with $\beta < 3/2$), the bias term $(A)$ dominates
191 the variance term $(B)$ and the value of $a^*$ changes accordingly. Conversely, when the variance term
192 $(B)$ is the dominant one (i.e., $\gamma_i(n) \leqslant cn^{-\beta}$ with $\beta > 3/2$), the threshold $a^*$ is governed by the
193 estimation uncertainty, being the bias negligible.

194 As common in optimistic algorithms for BAI (Audibert et al., 2010), setting a theoretically sound
195 value of exploration parameter $a$ (i.e., computing $a^*$), requires additional knowledge of the setting,
196 namely the complexity index $H_{1,\eta}(T)$.[9] In the next section, we propose an algorithm that relaxes this
197 requirement.

## 5 Phase-Based Algorithm: Rising Successive Rejects

199 In this section, we introduce the Rising Successive Rejects (R-SR), a phase-based solution
200 inspired by the one proposed by Audibert et al. (2010), which overcomes the drawback of R-UCBE of
201 requiring knowledge of $H_{1,\eta}(T)$.

202 **Algorithm** R-SR, whose pseudo-code is reported in Algorithm 2, takes as input the time budget $T$
203 and the number of arms $K$. At first, it initializes the set of the active arms $\mathcal{X}_0$ with all the available
204 arms (Line 1). This set will contain the arms that are still eligible candidates to be recommended.
205 The entire process proceeds through $K-1$ phases. More specifically, during the $j^{th}$ phase, the arms
206 still remaining in the active arms set $\mathcal{X}_{j-1}$ are played (Line 5) for $N_j - N_{j-1}$ times each, where:

$$N_j := \left\lceil \frac{1}{\overline{\log}(K)} \frac{T-K}{K+1-j} \right\rceil, \tag{11}$$

207 and $\overline{\log}(K) := \frac{1}{2} + \sum_{i=2}^{K} \frac{1}{i}$. At the end of each phase, the arm with the smallest value of the
208 pessimistic estimator $\hat{\mu}_i(N_j)$ is discarded from the set of active arms (Line 11). At the end of the
209 $(K-1)^{th}$ phase, the algorithm recommends the (unique) arm left in $\mathcal{X}_{K-1}$ (Line 13).

---

[9] We defer the empirical study of the sensitivity of $a$ to Section 7.

It is worth noting that R-SR makes use of the pessimistic estimator $\hat{\mu}_i(n)$. Even if both estimators defined in Section 3 are viable for R-SR, the choice of using the pessimistic estimator is justified by its better concentration rate $\mathcal{O}(n^{-1/2})$ compared to that of the optimistic estimator $\mathcal{O}(Tn^{-3/2})$, being $n \leqslant T$ (see Section 3).

Note that the phase lengths are the ones adopted by Audibert et al. (2010). This choice allows us to provide theoretical results without requiring domain knowledge (still under a large enough budget). An optimized version of $N_j$ may be derived assuming full knowledge of the gaps $\Delta_i(T)$, but, unfortunately, such a hypothetical approach would have similar drawbacks as R-UCBE.

**Bound on the Error Probability of** R-SR  The following theorem provides the guarantee on the error probability for the R-SR algorithm.

**Theorem 5.1.** *Under Assumptions 2.1 and 2.2, if the time budget $T$ satisfies:*

$$T \geqslant 2^{\frac{\beta+1}{\beta-1}} c^{\frac{1}{\beta-1}} \overline{\log}(K)^{\frac{\beta}{\beta-1}} \max_{i \in [\![2,K]\!]} \left\{ i^{\frac{\beta}{\beta-1}} \Delta_{(i)}(T)^{-\frac{1}{\beta-1}} \right\},  \tag{12}$$

*then, the error probability of* R-SR *is bounded by:*

$$e_T(\text{R-SR}) \leqslant \frac{K(K-1)}{2} \exp\left( -\frac{\varepsilon}{8\sigma^2} \cdot \frac{T-K}{\overline{\log}(K)H_2(T)} \right),$$

*where $H_2(T) := \max_{i \in [\![K]\!]} \left\{ i\Delta_{(i)}(T)^{-2} \right\}$ and $\overline{\log}(K) = \frac{1}{2} + \sum_{i=2}^{K} \frac{1}{i}$.*

Similar to the R-UCBE, the complexity of the problem is characterized by term $H_2(T)$ that, for the standard MAB setting, reduces to the $H_2$ term of Audibert et al. (2010). Furthermore, when the condition of Equation (12) on the time budget $T$ is satisfied, the error probability coincides with that of the SR algorithm for standard MABs (apart for constant terms). The following remark elaborates on the conditions of Equations (10) and (12) about the minimum requested time budget.

**Remark 5.1** (About the minimum time budget $T$). *To satisfy the $e_T$ bounds presented in Corollary 4.2 and Theorem 5.1,* R-UCBE *and* R-SR *require the conditions provided by Equations (10) and (12) about the time budget $T$, respectively. First, let us notice that if the suboptimal arms converge to an expected reward different from that of the optimal arm as $T \to +\infty$, it is always possible to find a finite value of $T < +\infty$ such that these conditions are fulfilled. Formally, assume that there exists $T_0 < +\infty$ and that for every $T \geqslant T_0$ we have that for all suboptimal arms $i \neq i^*(T)$ it holds that $\Delta_i(T) \geqslant \Delta_\infty > 0$. In such a case, the l.h.s. of Equations (10) and (12) are upper bounded by a function of $\Delta_\infty$ and are independent on $T$. Instead, if a suboptimal arm converges to the same expected reward as the optimal arm when $T \to +\infty$, the identification problem is more challenging and, depending on the speed at which the two arms converge as a function of $T$, might slow down the learning process arbitrarily. This should not surprise as the BAI problem becomes non-learnable even in standard (stationary) MABs when multiple optimal arms are present (Heide et al., 2021).*

# 6  Lower Bound

In this section, we investigate the complexity of the BAI problem for SRBs with a fixed budget.

**Minimum time budget T**  We show that, under Assumptions 2.1 and 2.2, any algorithm requires a minimum time budget $T$ to be guaranteed to identify the optimal arm, even in a deterministic setting.

**Theorem 6.1.** *For every algorithm $\mathfrak{A}$, there exists a deterministic SRB satisfying Assumptions 2.1 and 2.2 such that the optimal arm $i^*(T)$ cannot be identified for some time budgets $T$ unless:*

$$T \geqslant H_{1,1/(\beta-1)}(T) = \sum_{i \neq i^*(T)} \frac{1}{\Delta_i(T)^{\frac{1}{\beta-1}}}.  \tag{13}$$

Theorem 6.1 formalizes the intuition that any of the suboptimal arms must be pulled a sufficient number of times to ensure that, if pulled further, it cannot become the optimal arm. It is worth comparing this bound on the time budget with the corresponding conditions on the minimum time budget requested by Equations (10) and (12) for R-UCBE and R-SR, respectively. Regarding R-UCBE, we notice that the minimum admissible time budget in the small-$\beta$ regime is of order $H_{1,1/\beta}(T)^{\beta/(\beta-1)}$ which is larger than term $H_{1,1/(\beta-1)}(T)$ of Equation (13).[10]  Similarly, in the

---

[10] See Lemma D.12.

| | Error Probability $e_T(\cdot)$ | Time Budget $T$ |
|---|---|---|
| SRB | $\dfrac{1}{4}\exp\left(-\dfrac{8T}{\sigma^2 \sum_{i\neq i^*(T)}\frac{1}{\Delta_i^2(T)}}\right)$ | $\displaystyle\sum_{i\neq i^*(T)}\dfrac{1}{\Delta_i(T)^{\frac{1}{\beta-1}}}$ |
| R-UCBE | $2\,T\,K\,\exp\left(-\dfrac{a}{10}\right)$ | $\begin{cases}\left(c^{\frac{1}{\beta}}(1-2\varepsilon)^{-1}\left(\sum_{i\neq i^*(T)}\dfrac{1}{\Delta_i^{1/\beta}(T)}\right)+(K-1)\right)^{\frac{\beta}{\beta-1}} & \text{if } \beta\in(1,3/2)\\[2ex] \left(c^{\frac{2}{3}}(1-2\varepsilon)^{-\frac{2}{3}\beta}\left(\sum_{i\neq i^*(T)}\dfrac{1}{\Delta_i^{2/3}(T)}\right)+(K-1)\right)^{3} & \text{if } \beta\in[3/2,+\infty)\end{cases}$ |
| R-SR | $\dfrac{K(K-1)}{2}\exp\left(-\dfrac{\varepsilon}{8\sigma^2}\dfrac{T-K}{\overline{\log}(K)\max_{i\in[\![K]\!]}\left\{i\Delta_{(i)}^{-2}(T)\right\}}\right)$ | $2^{\frac{1+\beta}{\beta-1}}c^{\frac{1}{\beta-1}}\overline{\log}(K)^{\frac{\beta}{\beta-1}}\max_{i\in[\![2,K]\!]}\left\{i^{\frac{\beta}{\beta-1}}\Delta_{(i)}(T)^{-\frac{1}{\beta-1}}\right\}$ |

Table 1: Bounds on the time budget and error probability: lower for the setting and upper for the algorithms.

large-$\beta$ regime (i.e., $\beta > 3/2$), the R-UCBE requirement is of order $H_{1,2/3}(T)^3 \geqslant H_{1,2}(T)$ which is larger than the term of Theorem 6.1 since $1/(\beta-1) < 2$. Concerning R-SR, it is easy to show that $H_{1,1/(\beta-1)}(T) \approx \max_{i\in[\![2,K]\!]} i\Delta_{(i)}(T)^{-1/(\beta-1)}$, apart from logarithmic terms, by means of the argument provided by (Audibert et al., 2010, Section 6.1). Thus, up to logarithmic terms, Equation (12) provides a tight condition on the minimum budget.

**Error Probability Lower Bound** We now present a lower bound on the error probability.

**Theorem 6.2.** *For every algorithm $\mathfrak{A}$ run with a time budget $T$ fulfilling Equation (13), there exists a SRB satisfying Assumptions 2.1 and 2.2 such that the error probability is lower bounded by:*

$$e_T(\mathfrak{A}) \geqslant \frac{1}{4}\exp\left(-\frac{8T}{\sigma^2 H_{1,2}(T)}\right), \quad where \quad H_{1,2}(T) := \sum_{i\neq i^*(T)}\frac{1}{\Delta_i^2(T)}.$$

Some comments are in order. First, we stated the lower bound for the case in which the minimum time budget satisfies the inequality of Theorem 6.1, which is a necessary condition for identifying the optimal arm. Second, the lower bound on the error probability matches, up to logarithmic factors, that of our R-SR, suggesting the superiority of this algorithm compared to R-UCBE. Finally, provided that the identifiability condition of Equation (13), such a result corresponds to that of the standard (stationary) MABs (Audibert et al., 2010; Kaufmann et al., 2016). A summary of all the bounds provided in the paper is presented in Table 1.

## 7  Numerical Validation

In this section, we provide a numerical validation of R-UCBE and R-SR. We compare them with state-of-the-art bandit baselines designed for stationary and non-stationary BAI in a synthetic setting, and we evaluate the sensitivity of R-UCBE to its exploration parameter $a$. Additional details about the experiments presented in this section are available in Appendix G. Additional experimental results on both synthetic settings and in a real-world experiment are available in Appendix H.[11]

**Baselines** We compare our algorithms against a wide range of solutions for BAI:

- RR: uniformly pulls all the arms until the budget ends in a *round-robin* fashion and, in the end, makes a recommendation based on the empirical mean of their reward over the collected samples;
- RR-SW: makes use of the same exploration strategy as RR to pull arms but makes a recommendation based on the empirical mean over the last $\frac{\varepsilon T}{K}$ collected samples from an arm.[12]
- UCB-E and SR (Audibert et al., 2010): algorithms for the stationary BAI problem;
- Prob-1 (Abbasi-Yadkori et al., 2018): an algorithm dealing with the adversarial BAI setting;
- ETC and Rest-Sure (Cella et al., 2021): algorithms developed for the decreasing loss BAI setting.[13]

The hyperparameters required by the above methods have been set as prescribed in the original papers. For both our algorithms and RR-SW, we set $\varepsilon = 0.25$.

---

[11]The code to run the experiments is available in the supplementary material. It will be published in a public repository conditionally to the acceptance of the paper.

[12]The formal description of this baseline, as well as its theoretical analysis, is provided in Appendix E.

[13]This problem is equivalent to ours, given a linear transformation of the reward.

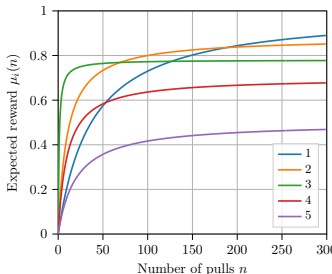 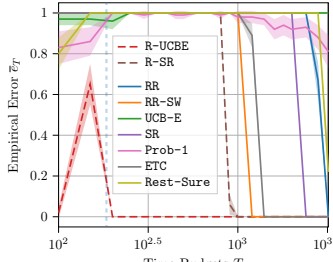 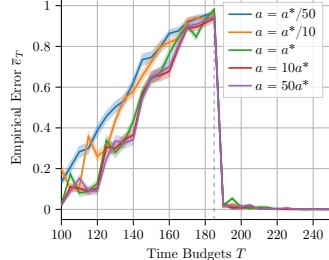

Figure 2: Expected values $\mu_i(n)$ for the arms of the synthetic setting.

Figure 3: Empirical error rate for the synthetically generated setting (100 runs, mean $\pm$ 95% c.i.).

Figure 4: Empirical error rate for the R-UCBE at different $a$ (1000 runs, mean $\pm$ 95% c.i.).

**Setting** To assess the quality of the recommendation $\hat{I}^*(T)$ provided by our algorithms, we consider a synthetic SRB setting with $K = 5$ and $\sigma = 0.01$. Figure 2 shows the evolution of the expected values of the arms w.r.t. the number of pulls. In this setting, the optimal arm changes depending on whether $T \in [1, 185]$ or $T \in (185, +\infty)$. Thus, when the time budget is close to that value, the problem is more challenging since the optimal and second-best arms expected rewards are close to each other. For this reason, the BAI algorithms are less likely to provide a correct recommendation than for time budgets for which the two expected rewards are well separated. We compare the analyzed algorithms $\mathfrak{A}$ in terms of empirical error $\overline{e}_T(\mathfrak{A})$ (the smaller, the better), i.e., the empirical counterpart of $e_T(\mathfrak{A})$ averaged over 100 runs, considering time budgets $T \in [100, 3200]$.

**Results** The empirical error probability provided by the analyzed algorithms in the synthetically generated setting is presented in Figure 3. We report with a dashed vertical blue line at $T = 185$, i.e., the budgets after which the optimal arm no longer changes. Before such a budget, all the algorithms provide large errors (i.e., $\overline{e}_T(\mathfrak{A}) > 0.2$). However, R-UCBE outperforms the others by a large margin, suggesting that an optimistic estimator might be advantageous when the time budget is small. Shortly after $T = 185$, R-UCBE starts providing the correct suggestion consistently. R-SR begins to identify the optimal arm (i.e., with $\overline{e}_T(\text{R-SR}) < 0.05$) for time budgets $T > 1000$. Nonetheless, both algorithms perform significantly better than the baseline algorithms used for comparison.

**Sensitivity Analysis for the Exploration Parameter of** R-UCBE We perform a sensitivity analysis on the exploration parameter $a$ of R-UCBE. Such a parameter should be set to a value less or equal to $a^*$, and the computation of the latter is challenging. We tested the sensitivity of R-UCBE to this hyperparameter by looking at the error probability for $a \in \{a^*/50, a^*/10, a^*, 10a^*, 50a^*\}$. Figure 4 shows the empirical errors of R-UCBE with different parameters $a$, where the blue dashed vertical line denotes the last time the optimal arm changes over the time budget. It is worth noting how, even in this case, we have two significantly different behaviors before and after such a time. Indeed, if $T \leqslant 185$, we have that a misspecification with larger values than $a^*$ does not significantly impact the performance of R-UCBE, while smaller values slightly decrease the performance. Conversely, for $T > 185$ learning with different values of $a$ seems not to impact the algorithm performance significantly. This corroborates the previous results about the competitive performance of R-UCBE.

# 8 Discussion and Conclusions

This paper introduces the BAI problem with a fixed budget for the Stochastic Rising Bandits setting. Notably, such setting models many real-world scenarios in which the reward of the available options increases over time, and the interest is on the recommendation of the one having the largest expected rewards after the time budget has elapsed. In this setting, we presented two algorithms, namely R-UCBE and R-SR providing theoretical guarantees on the error probability. R-UCBE is an optimistic algorithm requiring an exploration parameter whose optimal value requires prior information on the setting. Conversely, R-SR is a phase-based solution that only requires the time budget to run. We established lower bounds for the error probability an algorithm suffers in such a setting, which is matched by our R-SR, up to logarithmic factors. Furthermore, we showed how a requirement on the minimum time budget is unavoidable to ensure the identifiability of the optimal arm. Finally, we validate the performance of the two algorithms in both synthetically generated and real-world settings. A possible future line of research is to derive an algorithm balancing the tradeoff between theoretical guarantees on the $e_T$ and the chance of providing such guarantees with lower time budgets.

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
