# OpenReview forum: "Best Arm Identification for Stochastic Rising Bandits"
_NeurIPS.cc/2023/Conference — Submitted to NeurIPS 2023_

### Official Review · Reviewer_Sk8e · 2023-07-06

**Soundness:** 2 fair
**Presentation:** 3 good
**Contribution:** 2 fair
**Rating:** 5
**Confidence:** 3

**Summary:**

Stochastic Rising Bandits (SRBs) model sequential decision-making problems in which the expected reward of the available options increases after every time they are selected.
While previous works addressed the regret minimization problem, this paper studied the fixed-budget Best Arm Identification (BAI) problem for SRBs.
This work proposed the R-UCBE and R-SR algorithms and showed that these two algorithms with classical designs achieve a small failure probability when the time horizon is sufficiently large.
With a lower bound, the author(s) also showed that the R-SR algorithm is near-optimal and a sufficiently large horizon is unavoidable for any algorithm to perform well.
Lastly, experiments are provided to validate the empirical performance of BAI algorithms.


===========

The score is increased after I read the response from authors.

**Strengths:**

1. This work provides clearly formulates the rising bandit problem in Section 2.
2. Before presenting the R-UCBE and R-SR algorithms, the author(s) describe how to estimate the expected rewards of arms in Section 3, which actually imply the intuition of algorithm designs. This can inspire readers to propose efficient bandit algorithms even beyond this rising bandit setting.
3. In Section 7, experiments are provided to validate the empirical performance of BAI algorithms.

**Weaknesses:**

1. In Theorem 6.1, the lower bound on the time horizon $T$ depends on $\Delta_i(T)$. The quantity, $\Delta_i(T)$, depends on the instance, horizon $T$ and also the algorithm that we apply. Hence, $\Delta_i(T)$ seems to be a random variable, and I don't think a random variable should appear in the lower bound. Moreover, we usually expect a lower bound holds true for many algorithms, and the term $\Delta_i(T)$ seems to be different for different algorithms. or even different for the same algorithm in different trials.
2. I have a similar concern for the upper bounds on the failure probabilities of the R-UCBE and R-SR algorithms.

I think the contribution of this paper is much clearer, if my concerns above can be resolved.

**Questions:**

My major concerns as listed in the *Weakness* section. Some other suggestions are as below:
1. Line 45: 'failing to represent' may be revised to be 'but failed to represent'
2. At the bottom of Page 2, it states that 'A complete discussion of the related works is available in Appendix A. Additional motivating examples are discussed in Appendix B.' I think these are important components of the paper, and at least a brief version should be included in the main paper.

---

> ### Author Rebuttal · Authors · 2023-08-09
>
> We thank the Reviewer for the time spent reviewing our work and for the interesting comments. Below, we address the concerns of the Reviewer.
>
> ## Weaknesses
>
> > In Theorem 6.1, the lower bound on the time horizon $T$ depends on $\Delta_i(T)$. The quantity, $\Delta_i(T)$, depends on the instance, horizon $T$ and also the algorithm that we apply. Hence, $\Delta_i(T)$ seems to be a random variable, and I don't think a random variable should appear in the lower bound. Moreover, we usually expect a lower bound holds true for many algorithms, and the term $\Delta_i(T)$ seems to be different for different algorithms. or even different for the same algorithm in different trials.
>
> > I have a similar concern for the upper bounds on the failure probabilities of the R-UCBE and R-SR algorithms.
>
> We are happy to clarify the concern raised by the Reviewer. Given a specific time budget $T$ (which is an input of the fixed-budget BAI problem) and given an instance of the stochastic rising bandit (i.e., the expected rewards $\mu_i(t)$ functions), the values of the suboptimality gaps $\Delta_i(T) := \mu_{i^*(T)}(T) - \mu_i(T)$ are well-defined. They are (1) **algorithm-independent** (being defined through $T$ and $\mu_i(t)$ irrespective of the used algorithm) and (2) they are **not random variables** (since $T$ and $\mu_i(t)$ are deterministic). Thus, our lower bound (Theorem 6.1) depends on $\Delta_i(T)$ which, in turn, depends on the time budget $T$ and on the instance of the SRB (i.e., $\mu_i(t)$), as common in the BAI literature ([1], [2] and [3]). Indeed, **the lower bound holds for every algorithm run with a sufficiently large time budget (Theorem 6.1)**. Similarly, the upper bounds on the failure probabilities of the R-UCBE and R-SR algorithms depend on the same quantities. We will clarify these points in the final version of the paper.
>
> ## Questions
>
> > Line 45: 'failing to represent' may be revised to be 'but failed to represent'.
>
> We thank the Reviewer for pointing it out, we will adjust this sentence in the final version of the paper.
>
> > At the bottom of Page 2, it states that 'A complete discussion of the related works is available in Appendix A. Additional motivating examples are discussed in Appendix B.' I think these are important components of the paper, and at least a brief version should be included in the main paper.
>
> We will include these parts in the final version of the paper exploiting the additional page.
>
> ---
> [1] Audibert, Jean-Yves, Sébastien Bubeck, and Rémi Munos. "Best arm identification in multi-armed bandits." COLT. 2010.
>
> [2] Carpentier, Alexandra, and Andrea Locatelli. "Tight (lower) bounds for the fixed budget best arm identification bandit problem." Conference on Learning Theory. PMLR, 2016.
>
> [3] Kaufmann, Emilie, Olivier Cappé, and Aurélien Garivier. "On the complexity of best arm identification in multi-armed bandit models." Journal of Machine Learning Research 17 (2016): 1-42.

---

> > ### Comment · Reviewer_Sk8e · 2023-08-20
> >
> > Thanks for your clarification. The score is increased.

---

> > > ### Author Response · Authors · 2023-08-21
> > >
> > > We thank the Reviewer for taking the time to read our response. If the Reviewer has any other concerns about our work, we will be happy to clarify them.

---

### Official Review · Reviewer_2LCh · 2023-07-07

**Soundness:** 3 good
**Presentation:** 3 good
**Contribution:** 3 good
**Rating:** 6
**Confidence:** 3

**Summary:**

The paper studies the fixed-budget best arm identification (BAI) under the stochastic rising bandit (SRB) problem. The stochastic rising bandit is to assume that the mean reward will increase as one plays the arm more. By assuming a concave increasing reward, the paper provides upper bounds for two algorithms: R-UCBE (which is UCB-type) and R-SR (which is elimination-based). It further provides lower bound proof, and such a bound is matched by R-SR up to logarithmic factors. Numerical experiments are listed.


**Strengths:**

- The paper is well-written and carefully organized

- Matching lower and upper bounds (up to logarithmic factors)

**Weaknesses:**

1) Deterministic growth function $\gamma$, meaning that the randomness of $\gamma$ does not accumulate; Not the practical case (consider SGD, former parameters will affect the consecutive)

2) What if $c \rightarrow 0$? The requirement for $T$ (10) seems to vanish. Can you make further explanations?

Can you recover the bound of Audibert et al. (2010)?


**Questions:**

See above.

**Limitations:**

Na.

---

> ### Author Rebuttal · Authors · 2023-08-09
>
> We thank the Reviewer for the time spent on the review and for appreciating our work. Below, we address the concerns of the Reviewer.
>
> ## Weaknesses
>
> > Deterministic growth function $\gamma$, meaning that the randomness of $\gamma$ does not accumulate; Not the practical case (consider SGD, former parameters will affect the consecutive)
>
> We agree with the Reviewer that the proposed setting (motivated by SGD) is of practical interest. However, our work is framed in the established Stochastic Rising Bandit (SRB) setting of [1], which considers a noise model where the stochasticity in the observed reward does not accumulate. Indeed, considering the setting the Reviewer proposes (i.e., accumulating noise) would require to account for the *statistical dependence of consecutive observed rewards*. We agree that incorporating this accumulating noise model would make the setting closer to practical applications. However, we believe this would imply further technical challenges that would deserve a re-definition of the setting which, in our opinion, is out of the scope of the present paper.
>
> > What if $c \rightarrow 0$? The requirement for $T$ (10) seems to vanish. Can you make further explanations? Can you recover the bound of Audibert et al. (2010)?
>
> The term $c$ appears in Assumption 2.2, which requires a bound on the growth rates of the arms, i.e., $\gamma_i(n) \leq cn^{-\beta}$. Thus, when $c \rightarrow 0$, the arms have an expected reward that does not change with the number of pulls, i.e., $\gamma(t) \rightarrow 0$ for all $t \in [T]$. Setting $c\rightarrow 0$ has different effects on R-UCBE and R-SR.
> * **R-UCBE**: the minimum admissible time budget becomes $T \ge (K-1)^3$ (since $c\rightarrow 0$ we can freely select $\beta > 3/2$), depending on the number of arms $K$ only (Eq. 10). Our error probability bound becomes $2TK \exp \left(-\frac{\varepsilon^3}{40\sigma^2}\left(\frac{T^{1/3} - (K-1)}{H_{1, 2/3}(T)}\right)^3 \right)$ and does not correspond to that of [2]. This is because, differently from [2], we are using the *optimistic estimator* which involves the estimate of the increment, leading to looser concentration guarantees compared to the standard sample mean (see Lemma 3.2). A similar phenomenon is present in the original paper [1] for regret minimization, where the regret bound (Theorem 4.4) remains of order $T^{2/3}$ even for the stationary case, while $\sqrt{T}$ could be achieved by using the sample mean as an estimator. Note that using the *pessimistic estimator* in R-UCBE is not sound for the rising setting, but it is for the stationary setting, leading to guarantees analogous to that of [2].
> * **R-SR**: here, instead, the minimum admissible time budget requirement vanishes (Eq. 12 becomes simply $T>0$), as in [2]. Moreover, our error probability bound becomes $\frac{K(K-1)}{2}\exp\left( -{\color{black}{\frac{\varepsilon}{8 \sigma^2}}} \cdot \frac{T-K}{ \overline{\log} (K) H_2 }\right)$, matching the error probability of [2] apart from the constant term $\frac{\varepsilon}{8\sigma^2}$ deriving from the use of a windowed estimator (window of size $\lfloor \epsilon N_{i,t-1} \rfloor$) and the analysis based on $\sigma^2$-subgaussian rewards (instead of $[0,1]$-bounded rewards as in [2]).
>
> We will add a discussion on this in Sections 4 and 5 of the final version of the paper.
>
> ---
>
> [1] Metelli, A. M., Trovo, F., Pirola, M., & Restelli, M. Stochastic rising bandits. In International Conference on Machine Learning. 2022.
>
> [2] Audibert, J. Y., Bubeck, S., & Munos, R. Best arm identification in multi-armed bandits. In Conference on Learning Theory. 2010.

---

### Official Review · Reviewer_fA5g · 2023-07-07

**Soundness:** 3 good
**Presentation:** 4 excellent
**Contribution:** 3 good
**Rating:** 7
**Confidence:** 4

**Summary:**

The paper is about bandit best arm identification with fixed budget, in a non-stationary setting. This is a rested bandit problem: the mean reward of an arm changes each time it is pulled, but does not change when it is not pulled. The main assumption is that the mean reward is a non-decreasing, concave function of the number of pulls. Some results also use another assumption: an upper bound on the increments.
The authors introduce estimators of the mean rewards that are adapted to that setting and use them in two algorithms R-UCBE and R-SR, which are inspired by the UCBE and SR fixed budget algorithms. The paper contains upper bounds on the error probability of these algorithms as well as lower bounds on the error probability of any algorithm and a discussion of the minimal budget necessary to identify the best arm.

**Strengths:**

The rising bandit problem is important: it corresponds to allocating resources to different learning algorithms, in order to identify the one with best performance once fully trained.

The estimators and algorithms are well explained and motivated. The graphical representation of figure 1 is very helpful.

The lower and upper bounds show that the methods are close to optimal for the problem.

The discussion of the minimal budget necessary to identify the best arm is interesting and highlights a feature of rising bandits which is not present in standard BAI.

The experimental evaluation is convincing.

**Weaknesses:**

R-UCBE depends on a parameter that needs to be tuned using unavailable information, but that theoretical weakness is directly inherited from UCBE and the practical performance of the algorithm is very good. Hence this is a very mild weakness.

The lower bound of theorem 6.2 is of order $\exp(-T/H)$, while a famous feature of fixed budget BAI (in the stationary setting) is that this is not achievable. Indeed, it is shown in [Carpentier and Locatelli, Tight (lower) bounds for the fixed budget best arm identification bandit problem, Colt 2016] that there is a lower bound of order $\exp(-T/(H \log K))$, matching the upper bound of SR. In light of that lower bound, we would expect a similar result for rising bandits, stronger than Theorem 6.2.

**Questions:**

Is it possible to strengthen the lower bound?

Why nondecreasing and concave mean rewards? It should be easy to get large lower bounds if assumption 2.1 is not satisfied. Could you point to such results, in order to justify the assumption?

**Limitations:**

The limitations are adequately discussed. No concern about a negative societal impact.

---

> ### Author Rebuttal · Authors · 2023-08-09
>
> We thank the Reviewer for the time spent on the review and for appreciating our work. Below we provide the responses to the Reviewer's concerns.
>
> ## Weaknesses
>
> > R-UCBE depends on a parameter that needs to be tuned using unavailable information, but that theoretical weakness is directly inherited from UCBE and the practical performance of the algorithm is very good. Hence this is a very mild weakness.
>
> We agree with the Reviewer that this parameter is challenging to set in a theoretically-sound way, even if this problem, as the Reviewer noted, directly derives from UCBE. Aware of this issue, we conducted a **sensitivity analysis on parameter $a$** evaluating the error rate when we misspecify it. In Figure 4, we show that using values of $a$ with different magnitudes w.r.t. the optimal value $a^*$ still provides good performance for the R-UCBE. Noteworthy, R-SR is designed (as well as the original SR in the paper from [1]) with the goal to overcome the need for setting this parameter.
>
> > The lower bound of theorem 6.2 is of order $\exp(-T/H)$, while a famous feature of fixed budget BAI (in the stationary setting) is that this is not achievable. Indeed, it is shown in [Carpentier and Locatelli, Tight (lower) bounds for the fixed budget best arm identification bandit problem, Colt 2016] that there is a lower bound of order $\exp(-T/(H \log K))$, matching the upper bound of SR. In light of that lower bound, we would expect a similar result for rising bandits, stronger than Theorem 6.2.
>
> >Is it possible to strengthen the lower bound?
>
> We thank the Reviewer for raising this interesting point. We remark that our construction for the lower bound is based on the work [3] (Theorem 17) which presents a result compatible with ours (without the $\log K$ term). Nevertheless, we point out that our construction exploits the instances of Figure 5 and, specifically, the fact that the dissimilarity between the instances increases as the time $t$ increases. Consequently, our lower bound is derived for the simpler case in which they both achieved the "regime" behavior (derivation at line 713 when upper bounding the KL divergence at time $t$ with the one at time $T$). In other words, ours is a lower bound that holds for the stationary bandit we would have always considered $\mu_i(T)$ as the expected reward. Therefore, we believe that the proof of (Carpentier and Locatelli, 2016) can be integrated into our setting with no further challenges leading to the additional $\log K$ term. We will complement the lower bound analysis in the final version of the paper.
>
> ## Questions
>
> >Why nondecreasing and concave mean rewards? It should be easy to get large lower bounds if Assumption 2.1 is not satisfied. Could you point to such results, in order to justify the assumption?
>
> Without such assumptions (*non-decreasing* and *concave* expected rewards) the error probability cannot be guaranteed to be decreasing as a function of the budget $T$. From an intuitive perspective, this is similar to what happens for regret minimization in [2, Theorem 4.2], in which the authors demonstrate the non-learnability (i.e., an $\Omega(T)$ regret lower bound) when these two assumptions do not hold.
>
> From a technical perspective, we can easily show that the error probability no longer depends on $T$ when we just remove the **concavity assumption**.
> We consider two Gaussian bandits with unit variance. Let $\boldsymbol\nu$ be a 2-armed bandit with expected rewards $\mu_1(t) = 1/2$ and $\mu_2(t) = 3/4$, both $\forall t \in [T]$, thus its optimal arm is $i^*_{\boldsymbol\nu}(T)=2$ (we recall that the optimal arm in our setting is the one having the highest expected reward at $T$). Let $\boldsymbol\nu'$ be a 2-armed bandit with expected values $\mu_1(t) = 1/2 \; \forall t < T$, $\mu_1(t) = 1$ if $t=T$ and $\mu_2(t) = 3/4 \; \forall t \in [T]$, thus the optimal arm is $i^*_{\boldsymbol\nu'}(T)=1$. Notice that bandit $\boldsymbol\nu'$ violates the concavity assumption. Now, applying the **Bretagnolle-Huber** inequality, we have that:
>
> $$
> \begin{aligned}
> \\max\\{\\text{Pr}\_{\\boldsymbol\\nu}(\\hat{I}(T) \\neq 2), \\text{Pr}\_{\\boldsymbol\\nu'}(\\hat{I}(T) \\neq 1) \\}& \\ge \\frac{1}{4} \\exp\\left( -\\mathbb{E}\_{\\boldsymbol\\nu} \\left[\\sum\_{t=1}^T D\_{\\text{KL}}(\\boldsymbol\\nu\_{I\_t}(N\_{I\_t,t}), \\boldsymbol\\nu'\_{I\_t}(N\_{I\_t,t})) \\right]\\right)\\\\
> & \\ge \\frac{1}{4}  \\exp\\left( - D\_{\\text{KL}}(\\boldsymbol\\nu\_{1}(T), \\boldsymbol\\nu'\_{1}(T)) \\right) \\\\
> & = \\frac{1}{4}  \\exp\\left( - \\frac{1}{8} \\right),
> \end{aligned}
> $$
>
>
> where $\hat{I}(T)$ is the arm recommended at time $T$ and having observed that $D\_{\\text{KL}}(\\boldsymbol\\nu\_{I\_t}(N\_{I\_t,t}), \\boldsymbol\\nu'\_{I\_t}(N\_{I\_t,t})) = 0$ if $t < T$ regardless the arm $I\_t \in \\{1,2\\}$ and that  $D\_{\\text{KL}}(\\boldsymbol\\nu\_{I\_T}(N\_{I\_T,T}), \\boldsymbol\\nu'\_{I\_T}(N\_{I\_T,T})) \\le D\_{\\text{KL}}(\\boldsymbol\\nu_{1}(T), \\boldsymbol\\nu'_{1}(T)) = 1/8$. The last line shows a lower bound on the error probability that is budget-independent, thus, such a setting, obtained by removing the concavity assumption, is non-learnable. We will add this result in Section 6 of the final version of the paper.
>
> ---
>
> [1] Audibert, J. Y., Bubeck, S., & Munos, R. Best arm identification in multi-armed bandits. In Conference on Learning Theory. 2010.
>
> [2] Metelli, A. M., Trovo, F., Pirola, M., & Restelli, M. Stochastic rising bandits. In International Conference on Machine Learning. 2022.
>
> [3] Kaufmann, Emilie, Olivier Cappé, and Aurélien Garivier. "On the complexity of best arm identification in multi-armed bandit models." Journal of Machine Learning Research 17 (2016): 1-42.

---

### Official Review · Reviewer_uPrW · 2023-07-07

**Soundness:** 3 good
**Presentation:** 3 good
**Contribution:** 3 good
**Rating:** 6
**Confidence:** 2

**Summary:**

This study explores the stochastic rising bandits (SRB) in fixed-budget best arm identification (BAI). The authors initially formulate this novel problem setting and then introduce two types of estimators. For these estimators, they demonstrate upper bounds that match their lower bounds. Lastly, they validate the reliability of their approach through various experiments.






**Strengths:**

Firstly, this is my initial exposure to a paper on the SRB setting. Consequently, I'm not sure if the setting is truly novel. However, I am persuaded that the setting is both crucial and intriguing, particularly given the practical importance of the CASH problem. As I'm unable to gauge the novelty of the setting, my remarks are primarily oriented towards the technical aspects of BAI.

Firstly, the concepts of pessimistic and optimistic estimators are persuasive and well-founded. The authors deliver thorough and robust theoretical results for the estimators, consistent with established practices in this field. Even though the outcomes are not surprising, I see no strong grounds to reject this paper. On the whole, this study presents a typical analysis within an interesting and novel setting.

Please note, this is a preliminary review. I am currently delving into further details, including the proof. I may revise this review later.

**Weaknesses:**

It appears that a weakness resides in the need for a large budget. Moreover, verifying whether the condition is met could be challenging due to the somewhat complex form of the time budget constraint.

**Questions:**

Could we derive more practically applicable results for realistic settings by assuming specific models for rewards? For instance, in economics, we often presume particular models for a utility function that grows as the quantity of certain variables (e.g., consumption) increases. That is, if we define a certain mechanism (e.g., linear models) for the increasing rewards, could we achieve tighter results? If the authors' claims are correct, this study serves as a pioneering effort in this field. Therefore, incorporating such constraints could be seen as future work, and isn't necessarily a task the authors need to undertake at present. However, I am interested in understanding the potential for such an extension.

**Limitations:**

See weakness.

---

> ### Author Rebuttal · Authors · 2023-08-09
>
> We thank the Reviewer for the time spent on the review and for appreciating our work. Below we provide the responses to the Reviewer’s concerns.
>
> ## Weaknesses
>
> > It appears that a weakness resides in the need for a large budget. Moreover, verifying whether the condition is met could be challenging due to the somewhat complex form of the time budget constraint.
>
> A large time budget $T$ is required to allow the arms to almost reach their "regime" values. Indeed, even if in practice our algorithms perform well for way smaller time budgets (as shown in the experiments of Section 7), the theoretical guarantees consider the worst-case scenarios. We remark that a minimum time budget $T$ is indeed **unavoidable** as proved by our lower bound for the time budget (Theorem 6.1) which is matched (up to logarithmic terms) by our R-SR algorithm (Theorem 5.1).
>
> ## Questions
>
> > Could we derive more practically applicable results for realistic settings by assuming specific models for rewards? For instance, in economics, we often presume particular models for a utility function that grows as the quantity of certain variables (e.g., consumption) increases. That is, if we define a certain mechanism (e.g., linear models) for the increasing rewards, could we achieve tighter results? If the authors' claims are correct, this study serves as a pioneering effort in this field. Therefore, incorporating such constraints could be seen as future work, and isn't necessarily a task the authors need to undertake at present. However, I am interested in understanding the potential for such an extension.
>
> As the Reviewer noted, we address the setting in which the only assumptions relate to the (1) *non-decreasing* and (2) *concave* shape of the expected values, which are, in a sense, the less demanding ones (see also the Response to Reviewer fA5g for a discussion on the need for these assumptions). We agree with the Reviewer that considering particular functional forms of the expected rewards $\mu_i(t)$ (i.e., beyond (1) and (2)), proper of specific realistic settings (e.g. economics), will likely lead to tighter and more applicable results. In principle, one could consider a generic **known parametric functional form** for the expected rewards $\mu_{i}(t;\boldsymbol\theta)$ depending on an **unknown vector of parameters** $\boldsymbol\theta$, making use of suitable estimators and exploiting the uncertainty on the $\boldsymbol\theta$ estimate. A specific example of this is [1] in which a particular known polynomial class of functions is considered for the expected rewards $\mu_{i}(t;\boldsymbol\theta)$. We will add a comment on this in Section 8 of the final version of the paper.
>
> ---
>
> [1] Cella, L., Pontil, M., & Gentile, C. Best model identification: A rested bandit formulation. In International Conference on Machine Learning. 2021.

---

### Decision · Program_Chairs · 2023-09-21

**Decision:**

Reject

**Comment:**

The paper studies the problem of best arm identification in the so-called stochastic rising bandits (SRB). In the latter, the average rewards increase as they are selected, but in a controlled manner -- non-decreasing and concave, and with increment vanishing faster than $n^{-\beta}$ for $\beta>1$. The notion of best arm is chosen as the arm that would be the best should all the $T$ samples had been allocated to this arm. The authors propose two algorithms, namely R-UCBE and R-SR, that extend their respective counterparts in classical stochastic MABs. The performance analysis of these algorithms relies on concentration results for pessimistic and optimistic estimators (Lemmas 3.1 and 3.2). The analysis is well conducted but arguably not very novel.

The paper was positively received by the reviewers. However, several aspects of the paper must be improved so as to clarify its contributions.

-  The most important aspect is to put the paper's contributions considering existing papers. SRB are not new and have received some attention recently. The authors should do a much better comparison to existing work, and in particular with (Metelli et al. , 2022) and (Cella et al. 2021). The former indeed deals with regret minimization but considers the same setting (increasing concave rewards) as this submission. It seems however that some of the techniques used in (Metelli et al 2022) are used here (is it the case for the estimators? See Fig. 1 in both papers). The authors need to clarify what new techniques they introduce here.
For the latter (Cella et al. 2021), indeed they assume that the rewards evolve as $a+b/n^\beta$ for some unknown parameters $a$ and $b$, whereas the present submission considers this rate as an upper bound of the rewards. This improvement is not spectacular. The related work section can be found in Appendix, but is too limited to understand whether the contributions of the present submission are technically novel.

- The reader may wonder why, in the setting where the rewards of the arms evolve over time, best arm identification (BAI) is an interesting task. It is actually difficult to define the best arm: here the authors decided to define the best arm as that with the largest expected reward should all the $T$ samples had been allocated to this arm. Why is it a good definition? The authors could motivate this definition. When looking at the use cases proposed by the authors, one may argue that the performance metric presented in (Cella et al. 2021) is more appropriate, see Eq (2) there. This metric is some kind of simple regret (we do not care whether the best arm is identified if the reward of the chosen arm is very close to that of the best arm).

- Although the reviewers liked problem setting, the authors could do a better job at motivating the model. The CASH problem seems interesting, but the authors could develop this example more in detail in the paper. For example, they could use it in Section 7 when presenting numerical experiments.

- It seems that Assumption 2.2 is critical when analyzing the performance of the algorithms. Can the authors explain what would be the problem when relaxing this assumption?

- For the experiments presented in Section 7, the authors seem to have decided to illustrate the performance of their algorithms on a toy example. Presenting an example here with real data (as done in Appendix H) would be much more convincing.

The paper studies an interesting problem, but the authors could make its contributions more convincing by 1. really motivating why BAI is an interesting task in SRB, 2. highlighting its technical novelty.